# Towards Agnostic and Holistic Universal Image Segmentation with Bit Diffusion

Jakob Lønborg Christensen[1], Morten Rieger Hannemose[1], Anders Bjorholm Dahl[1], and Vedrana Andersen Dahl[1]

[1]Technical University of Denmark
{jloch, mohan, abda, vand}@dtu.dk

## Abstract

This paper introduces a diffusion-based framework for universal image segmentation, making agnostic segmentation possible without depending on mask-based frameworks and instead predicting the full segmentation in a holistic manner. We present several key adaptations to diffusion models, which are important in this discrete setting. Notably, we show that a location-aware palette with our 2D gray code ordering improves performance. Adding a final tanh activation function is crucial for discrete data. On optimizing diffusion parameters, the sigmoid loss weighting consistently outperforms alternatives, regardless of the prediction type used, and we settle on x-prediction. While our current model does not yet surpass leading mask-based architectures, it narrows the performance gap and introduces unique capabilities, such as principled ambiguity modeling, that these models lack. All models were trained from scratch, and we believe that combining our proposed improvements with large-scale pretraining or promptable conditioning could lead to competitive models.

## 1 Introduction

In universal image segmentation, the goal is to segment images from many data modalities with a single model. Conversely, narrow image segmentation is characterized by specializing on a single dataset or task, such as brain tumor segmentation. In recent times, the image segmentation field has favored mask-based segmentation models such as Mask-RCNN [1] and the Segment Anything Model (SAM) [2, 3]. These foundation models are used as general problem solvers that can be finetuned or prompted for narrow downstream tasks.

Universal segmentation systems increasingly face two, often competing, requirements: agnostic behavior and a holistic view of the images. By *agnostic* we mean the ability to segment objects without relying on a fixed label set. Agnostic models focus on masks while unbound by labels, enabling the model to generalize across domains and unseen categories. By *holistic* we mean a model that considers

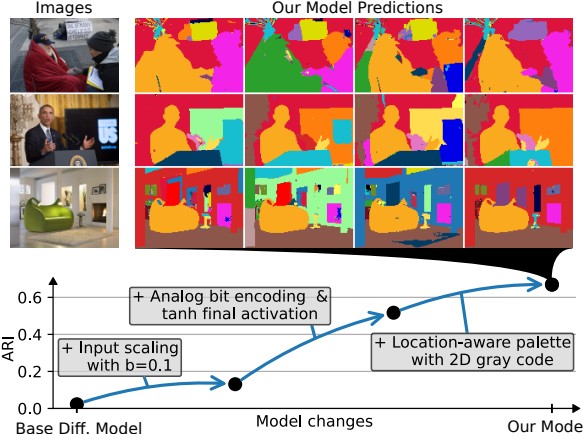

**Figure 1.** The modifications to a base diffusion model and their performance gains, visualized along with samples from our model.

the whole image when producing segmentations, including inter-mask correlations. I.e., choosing the same semantic division for separate masks. In practice, the first property enables open-world and cross-dataset use, while the second reduces segmentation inconsistencies.

We study diffusion-based segmentation as a route to achieve these goals. Diffusion models are well known for revolutionizing image generation [4], but in our setting the image is only a conditional input to the task of generating the segmentation. Additionally, using diffusion models makes ambiguity modeling possible.

Direct diffusion over discrete, high-dimensional label spaces is non-trivial. Diffusion was developed with continuous targets in mind, and it therefore faces multiple challenges when dealing with discrete data such as segmentations. Our approach combines various ideas from the diffusion research landscape. The addition of these ideas is essential to raise our model's performance. Our main contribution is to adapt the following existing techniques to work for universal diffusion segmentation (see Fig. 1), and improving them with our novel additions:

1. **Input scaled noise schedule [5].** Like [6], we use an input-scaled [5] diffusion noise schedule, in order to make the denoising problem suitably

Proceedings of the 7th Northern Lights Deep Learning Conference (NLDL), PMLR 307, 2026.

hard for discrete target spaces and improving training stability for segmentation.

2. **Analog bit diffusion encoding [7].** We encode $2^k$ classes with $k$ signed bits and train the diffusion model to predict bit-valued targets, reducing dimensionality while preserving a simple route back to class indices. We suggest adding a $\tanh(\cdot)$ activation, as it aligns the network's outputs with the discrete bit codes and yields better-calibrated probabilities.

3. **Location-aware palette [8] (LAP).** We adapt an LAP to reduce the downsides of the analog bit encoding when paired with our ordering that follows a 2D gray code. The LAP assigns indices by mask location, creating consistent targets in an agnostic setting, improving training.

## 2 Related Works

The most common flavor of universal segmentation models are mask-based (e.g. Mask R-CNN [1]). They generally work by detecting candidate regions for potential masks, and then handling each candidate separately as a binary mask prediction and/or classification problem [9–12]. Promptable class-agnostic systems such as SAM [2] demonstrate strong open-world mask extraction, but are still relying on binary foreground/background mask prediction. Masks are produced independently across the image and are therefore not holistic. An ideal universal segmentation model should be holistic, to avoid inconsistency when producing e.g. repeating objects in an image or simply to avoid overlapping masks.

We observe that mask-based models are limited to predicting one mask at a time because they optimize for mean predictions. For full agnostic segmentations, the mean would deviate too far from any ground truth due to scene uncertainty. This issue is less severe for binary masks, where variance is low, and absent in non-agnostic models with fixed vocabularies. Traditional losses such as cross-entropy or Dice push toward single estimates even when boundaries are ill-defined or annotators disagree, often blurring details and under-representing multi-modal solutions. Probabilistic segmentation explicitly models these uncertainties, e.g., Probabilistic U-Net and its variants [13–15], and hierarchical variational approaches [16]. Bayesian [17] and ensemble-style methods estimate uncertainty but often at a significant compute cost or weaker distributional guarantees. Diffusion-based segmentation offers a generative alternative that can sample diverse, plausible masks and produce uncertainty maps by construction [18–20]. Previously mentioned generative models all operate on narrow tasks instead of universal segmentation.

Another diffusion-based method, pix2seq-D [6] focused on panoptic segmentation with diffusion models. They took advantage of the ambiguity modeling inherent to generative models by splitting semantic masks into instance masks without running into combinatorial problems. Their method also made use of input scaling and analog bits, to deal with the discrete data domain.

The paper Unified Representation for Image Generation and Segmentation (UniGS) [8] is the most comparable to our approach, as it also tackles universal image segmentation with diffusion models. UniGS treats masks and images within a single latent-diffusion framework by representing entity-level masks as RGB colormaps aligned to the image domain. They choose the RGB space because their network is a finetuned Stable Diffusion [4] model (text-to-image). Decoding masks from the predicted RGB encoding is tricky, requiring the introduction of a progressive dichotomy module. The authors also introduce a location-aware color palette that assigns consistent colors to entities based on spatial location. Relative to UniGS, our work only targets the segmentation domain and instead of utilizing a pretrained model such as Stable Diffusion, we train from scratch. Training from scratch comes with upsides and downsides, namely we are restricted to working at a small scale but we are able to study the properties of the model in an unbiased setting, and without restrictions on modeling choices.

## 3 Methods

### 3.1 Diffusion Model

We use a continuous time diffusion model [21, 22] ranging from time $t = 0$ (data) to $t = 1$ (noise). The diffusion sample $\mathbf{x}_t$ is given by the equation

$$\mathbf{x}_t = \alpha(t)\mathbf{x}_0 + \sigma(t)\epsilon, \qquad (1)$$

where $\mathbf{x}_0$ is data, $\epsilon$ is i.i.d unit Gaussian noise. The functions $\alpha(t)$ and $\sigma(t)$ are the data and noise coefficients, respectively. For a diffusion segmentation model such as ours, the data is a segmentation map. The image is a conditional input which we concatenate across the channel dimension. The model operates in pixel space, since recent research shows these models can be competitive latent diffusion alternatives [23, 24].

In order to predict $\mathbf{x}_0$, the network can predict it directly ($x$-prediction), predict the noise ($\epsilon$-prediction), or predict $\mathbf{v} = \alpha(t)\epsilon - \sigma(t)\mathbf{x}_0$ ($v$-prediction [21]). Each of these predictions parameterize the others based on Eq. (1).

We employ a convolutional neural network (CNN) with an attention mechanism to predict the mean of

the conditional distribution $p(\mathbf{x}_0|\mathbf{x}_t)$, i.e. predicting the data from a noisy latent sample. Based on [22], the model can generate segmentation maps by denoising pure noise into segmentation maps over a number of timesteps. Sampling refers to turning random gaussian noise into a prediction. We always use equidistant timesteps from $t = 1$ to $t = 0$ when sampling.

The model is trained with the weighted MSE loss function [22]

$$L(\mathbf{x}) = \mathbb{E}_{t \sim \mathcal{U}(0,1)} \left[ w(t) \| \mathbf{x}_0 - \hat{\mathbf{x}} \|^2 \right], \qquad (2)$$

where $\hat{\mathbf{x}} = \hat{\mathbf{x}}_\theta(\mathbf{x}_t, t)$ is the neural network prediction of the data, $\mathbf{x}_0$. The loss weighting, $w(t)$, can emphasize the importance of different parts of the diffusion process, and following [23, 24] we use the sigmoid loss weighting with a bias of $-4$.

## 3.2 Bit Diffusion

Analog Bit Diffusion [7] is a modification to diffusion models that enable the model to work with high-dimensional discrete data, while maintaining a low dimensional latent space. Instead of representing discrete data as e.g. one-hot vectors, we represent the $2^k$ classes as $n_{\text{bits}} = k$ bits. We use $2^6 = 64$ classes corresponding to $n_{\text{bits}} = 6$. Negative bits have a value of $-1$ instead of 0, to make their distribution zero-mean and unit variance.

The diffusion process works in the bit space, and can be easily converted to the class space by thresholding the bits at 0 and converting from the binary representation.

With the bit diffusion formulation, the model should only predict values within $[-1, 1]$ with heavy emphasis on the endpoints of the interval. The $\tanh(\cdot)$ activation function is well suited for such a distribution, and we therefore apply it as a final activation (in cases where the model predicts the data directly). The non-thresholded bit activations enable conversion to a direct probability map. Let $\hat{y}$ be the predicted bits for some pixel. The probability that the pixel has the binary sequence $y$ is given by

$$p(y|\hat{y}) = \prod_{i=0}^{n_{\text{bits}}-1} p(y_i|\hat{y}_i) = \prod_{i=0}^{n_{\text{bits}}-1} \left( 1 - \frac{|y_i - \hat{y}_i|}{2} \right). \tag{3}$$

The equation above makes the downside of using a bit encoding clear. It does not model correlations between bits, but instead considers each bit probability separately. In reality, the bits are often correlated and as the correlation grows the bit encoding becomes less accurate.

## 3.3 Noise Schedule and Input Scaling

The noise schedule is parameterized by $\gamma : [0, 1] \to [0, 1]$, a monotonically decreasing function. We use

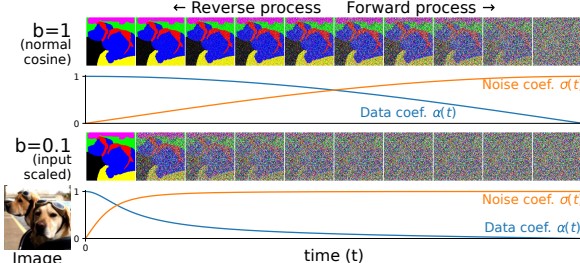

**Figure 2.** The cosine noise schedule with latent diffusion samples $x_t$ for various values of $t$. The latent samples use 3 bits (up to 8 masks) to make them viewable as RGB images.

a variance preserving noise schedule, where the coefficients are given by

$$\alpha(t) = \sqrt{\gamma(t)}, \quad \sigma(t) = \sqrt{1 - \gamma(t)}. \tag{4}$$

The variance preserving property enables parameterizing both set of coefficients with a single function. A common choice for the noise schedule is the cosine schedule, which is given by $\gamma(t) = \cos(t\pi/2)^2$ .

Consider the upper row of latent samples in Fig. 2. As a consequence of using discrete data with high spatial correlation, it is easy to reconstruct the data for large parts of the diffusion process. If the model is able to only consider the latent sample for large parts of the diffusion process during training, then the resulting model will be poor since it ignores the image during inference. The issue stems from the fact that the noise schedule is too easy, i.e. it can become trivial to reconstruct the data.

To address these concerns we use input scaling [5], which can be used to make diffusion noise schedules harder. Input scaling was originally introduced to deal with large images since increasing the number of pixels lessens the effect of the noise. The idea behind input scaling is to make noise schedule harder by lowering the signal-to-noise ratio (SNR). The SNR is given by

$$\text{SNR}(t) = \frac{\alpha(t)}{\sigma(t)} = \frac{\sqrt{\gamma(t)}}{\sqrt{1 - \gamma(t)}}, \tag{5}$$

and is lowered by multiplying with some constant $b \in [0, 1]$, called the input scale. One can show that solving

$$\frac{\sqrt{\gamma_b(t)}}{\sqrt{1 - \gamma_b(t)}} = b \frac{\sqrt{\gamma(t)}}{\sqrt{1 - \gamma(t)}}, \tag{6}$$

for the input scaled noise schedule, $\gamma_b(t)$, yields the expression

$$\gamma_b(t) = \frac{b^2 \gamma(t)}{(b^2 - 1)\gamma(t) + 1}. \tag{7}$$

Thus, all equations involving the noise schedule can be reused, except by replacing the original $\gamma(t)$ with the input scaled $\gamma_b(t)$.

## 3.4 Location-aware Palette

The segmentation model is class-agnostic, and therefore the class numbers which we assign objects can be permuted without changing the task. A valid option is thus to assign random class numbers, but a better option is a location-aware palette (LAP)[8]. With an LAP, each mask is assigned a class number based on the mask centroid. An $L \times L$ grid is constructed across the image, with each square associated with a class number. When multiple mask centroids share a grid, they are instead given the class number of the nearest free grid square. Without an LAP, the best prediction at $t = 1$ is a zero-image, since the data is pure noise and the expected value of random bits is zero. When the prediction is independent of the image, there is no useful learning signal. When using an LAP, classes are biased towards the nearby LAP class indices, thus providing a learning signal for the parts of the diffusion process where the latent sample is largely noise.

The analog bit encoding has difficulty representing class distributions with multiple classes when the bits of the classes differ significantly (see supplementary material for details). By exploiting the LAP, we can increase the likelihood of adjacent class regions sharing their bit encoding digits. To this end, we arrange the bit codes in the $L \times L$ grid as a 2-dimensional gray code [25]. This ensures each 1-connectivity pair of neighbors only differ by 1 bit in the LAP. Since we use $n_{\text{bits}} = 6$ we have $L = \sqrt{2^6} = 8$.

## 4 Experiments

### 4.1 Evaluation Setup

As a basis for our experiments we use the Entity-Seg [26] dataset, consisting of 33,227 images each fully segmented with high-quality agnostic class labels across a variety of modalities. We partition the dataset on a holdout basis with an 80-10-10 split (train-val-test) and we use a $128 \times 128$ resolution version of their dataset using the padding strategy from [2]. Our model is a 38.5m parameter attn-UNet [23, 24] trained for 300k iterations with a batch size of 8. The learning rate was set at $1e-4$, with linear warmup for the first 1000 iterations and decreased with a cosine schedule for the last 50k iterations. We used the AdamW [27] optimizer.

We compare quantitatively using two metrics. The first is the adjusted rand index (ARI), which is based on the probability of two random pixels agreeing in the ground truth and prediction on whether they should belong to the same class or different classes. The adjusted formulation ensures the expected value for a random prediction is 0 while still keeping a perfect prediction at a score of 1. The

|          | No LAP |       | w/ LAP |       |
|----------|--------|-------|--------|-------|
| **Encoding** | **ARI** | **IoU** | **ARI** | **IoU** |
| Onehot   | 0.168  | 0.186 | 0.528  | 0.323 |
| RGB      | 0.460  | 0.283 | 0.524  | 0.312 |
| Analog Bits | **0.515** | **0.368** | **0.670** | **0.432** |

**Table 1.** Performance for models trained with different encoding types.

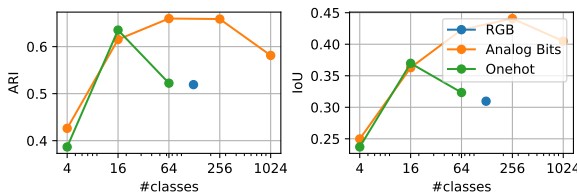

**Figure 3.** Performance for the three encoding types as the number of representable classes are varied.

second metric is the Intersection over Union (IoU) matched with the Hungarian algorithm. Following [26] we only compute the mean over non-empty ground truth classes after matching ground truths with predictions.

Our main model uses $x$-prediction and the sigmoid loss weights. The noise schedule is a cosine noise schedule with input scale parameter $b = 0.1$. The training data class indices are chosen based on a location-aware palette (LAP) that promotes similar analog bit encodings. A final activation function of $\tanh(\cdot)$ is applied to the network. For sampling, we use 8 timesteps and a guidance weight of 1.0 unless otherwise is stated.

### 4.2 Comparisons

We compare our model with the onehot and RGB encodings. The results (shown in Table 1) show that our model using analog bits improves upon the alternatives. The contrast is especially large when the models are trained with an LAP.

The analog bit encoding has exponential efficiency in the number of classes it can represent, which is clear when comparing how many classes the methods can represent in Fig. 3. Onehot and analog bits are similar in performance until around 16 classes when onehot falls off. We use 64 classes as a baseline for the rest of the experiments, since 96.14% of images in the dataset have $<= 64$ objects.

There is still a significant gap between our model and SOTA agnostic segmentation models (see Table 2). The mask-based models such as Mask2Former and CropFormer are more consistent despite not having a holistic segmentation pipeline.

Our model was trained with an empty image in 5% of training samples, as it enables using classifier free guidance [28] during sampling to increase the conditioning strength. To optimize sampling,

| | ARI | IoU | #Params |
|---|---|---|---|
| SAM base[2] | 0.478 | 0.467 | 93.7m |
| Mask2Former[11] | 0.852 | 0.663 | 47.4m |
| CropFormer[26] | **0.856** | **0.676** | 49.0m |
| Ours | 0.672 | 0.438 | 38.5m |

**Table 2.** Performance comparison with SOTA models on the public validation set. This validation set was a subset (roughly 4%) of the 10% of data we used a test set data.

we vary the guidance weight (gw) and number of sampling timesteps (see Fig. 4 and Fig. 5). We see that around only 8 sampling steps is optimal and performance only degrades slightly when using more steps. Based on the ARI metric $gw = 1.0$ is best, while IoU prefers a stronger $gw = 2.5$. Note that $gw = 0.0$ is the same as normal sampling with no guidance.

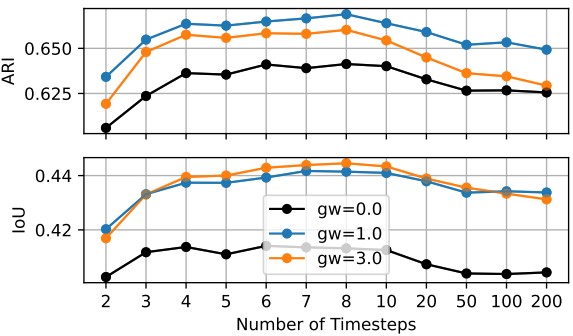

**Figure 4.** Mean performance on the validation set as the number of timesteps is varied for different guidance weights (gw).

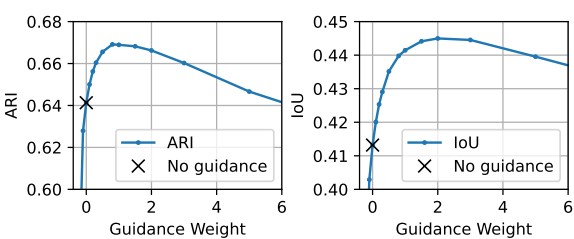

**Figure 5.** Mean performance on the validation set as the guidance weight is varied.

We increase the number of samples for each image in Fig. 6 to see the potential gains if one had an oracle to select the best prediction. More realistically this indicates the usefulness of a human in the loop or a test time augmentation (TTA) heuristic to select or aggregates samples. Using a larger guidance weight comes with a small penalty for the sample diversity as we see a smaller gain in performance.

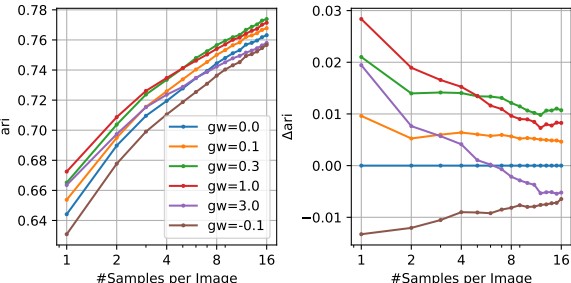

**Figure 6.** Mean performance when selecting the best segmentation from multiple samples. Shown in absolute ARI (left) and relative to no guidance (right).

## 4.3 Ablations

To investigate the best pair of prediction type and loss weights, we train a range of models while varying the available options. The results are seen in Fig. 7. With all prediction types, the sigmoid loss weights perform the best. The model with $\epsilon$-prediction is slightly better than $x$-prediction. However, when inspecting samples produced by the model (see Fig. 8, the $\epsilon$-prediction often failed to remove all the noise. One might think thresholding would solve this problem, but based on qualitative inspection of samples it seems the denoising trajectory is affected, leaving small noisy patches of nonsensical labels. A much more visible symptom of the same effect is visible for the model with no tanh activation. Given the tiny difference in performance, we therefore still use $x$-prediction.

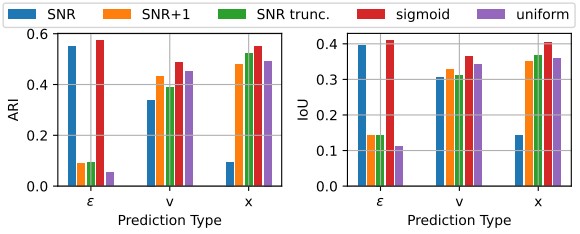

**Figure 7.** Mean performance for models trained with different prediction types and loss weights. These models were trained without LAP and $b = 0.1$.

The LAP encoding setup described in Section 3.4 is the one we call **similar**, since adjacent encodings are similar. Additionally, we also consider an LAP with **random** class indices and one which maximizes the **difference** of adjacent classes based on a greedy heuristic. The results in Table 3 show that in all cases, an LAP significantly increases performance. Additionally, the more similar the bit encodings of adjacent class indices, the better the performance.

To study the effect of input scaling we train a variety of models while varying $b$ (see Fig. 9). A value of $b = 0.1$ is close to optimal for our application. Note that $b = 1.0$ corresponds to a model with no

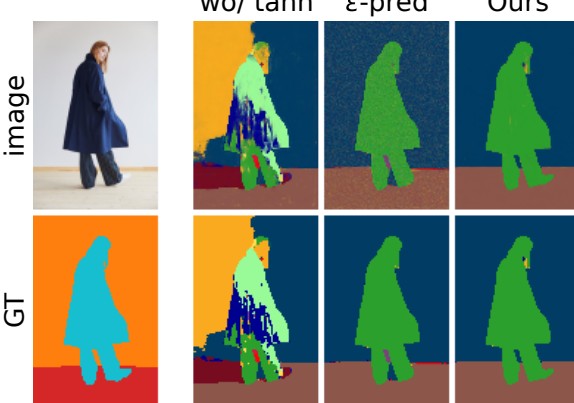

**Figure 8.** A qualitative example to illustrate the difference in samples produced by a model without $\tanh(\cdot)$ activation and with $\epsilon$-prediction, compared to our model.

| LAP | None | Different | Random | Similar |
|-----|------|-----------|--------|---------|
| **ARI** | 0.517 | 0.640 | 0.644 | **0.670** |
| **IoU** | 0.367 | 0.422 | 0.421 | **0.434** |

**Table 3.** Mean performance for models trained with different types of LAP.

input scaling. The average metrics are more than doubled by just adding input scaling to the noise schedule.

## 5   Discussion

Our experiments show that analog bits consistently outperforms RGB and one-hot encodings in agnostic segmentation. The relative gains are largest when class indices are assigned with a location-aware palette (LAP). We theorize that the gain in performance is an effect of an improved training process. Previously the network would learn little to nothing near $t = 1$, just producing a zero-mean prediction, but the bias from the LAP lets it encode segmentations at any timestep. By ordering bit codes of the LAP with a 2D gray ordering, we reduced differences between neighboring bits (the similar model). This allowed the model to express soft ambiguity between adjacent masks without paying the penalty of spreading probability mass over many unrelated

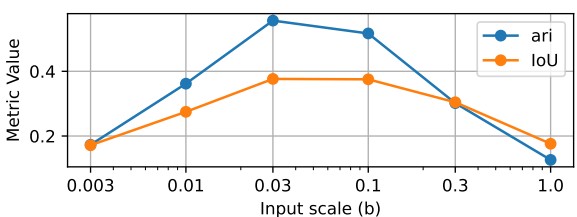

**Figure 9.** Performance for non-LAP models when varying the input scale parameter ($b$).

codes.

The analog bit encoding was preferred in our networks that were trained from scratch. An interesting research question is whether the same holds for tasks similar to that of UniGS [8]. The UniGS model was designed with the RGB encoding specifically because stable diffusion operates in RGB space. It may be possible to add a head to the segmentation branch to make this conversion possible. Given UniGS already reports competitive scores in segmentation benchmarks, replacing RGB colormaps with analog bits could perhaps push the unified generator–segmenter model to the forefront.

We framed our model as coming from successive additions of first the input scaled noise schedule then analog bits + tanh and finally the LAP. It is possible to add these model enhancements in a different order. We used the most impactful additions first, meaning an input scaled noise schedule was the most effective in improving training and performance. Adding analog bits or an LAP before input scaling would lead to less improvement because these methods needed the stability offered by input scaling before they could shine.

The best results were achieved when the network used $x$-prediction. Across prediction types ($x$, $v$, and $\epsilon$), the sigmoid loss weighting dominates alternatives, provided its bias is tuned. In our early experiments, we found a bias of $-4$ to be effective. Input scaling makes the schedule "hard enough" for discrete targets: reducing the effective SNR with $b \approx 0.1$ more than doubles ARI over the unscaled cosine schedule. Since input scaling was introduced in order to tackle the problem of high-dimensional spatial data, one can expect it should be lowered further than $b = 0.1$ for models with larger image sizes than $128 \times 128$.

We observe that only $\sim$8 denoising steps are sufficient for near-optimal performance, with modest degradation beyond that. Typically, diffusion models using the basic DDPM [29] sampler require many hundreds of steps for decent results, but discrete data may have lowered it. It is unclear to us why the model performance degrades with more steps. Further research is needed and perhaps there is some performance to be gained by preventing this collapse.

We find similar classifier-free guidance values as those commonly used for text-to-image models. A guidance weight around 1 seems to help conditioning without collapsing diversity. We only explored image guidance, but future work could extend to other promptable signals such as weak supervision (points, boxes, scribbles), class labels, or few-shot examples. Modern universal segmentation systems must be promptable to be useful in practical settings. The ability to control condition strength on these inputs would provide a whole new dimension to promptable

segmentation that traditional non-diffusion models do not have.

Overall, we provide a concrete path to make diffusion models viable for universal segmentation: analog bit diffusion for discrete labels, a noise schedule with input scaling, LAP for agnostic supervision, and a robust loss weighting. These choices yield consistent gains and make the method competitive in agnostic/holistic settings. At the same time, in broad foundation scenarios dominated by mask-classification architectures, our current model does not yet surpass strong discriminative baselines such as MaskFormer/Mask2Former or promptable SAM variants [2, 10, 11]. This gap likely reflects scale (data, compute, pretraining) and it motivates future work based on pretrained networks.

## 6    Conclusion

Diffusion models can serve as a viable framework for universal segmentation when adapted to discrete labels. It is necessary to modify the model to suit the discrete domain. Analog bits prove to be an effective encoding scheme, combined with a 2D gray code location-aware palette. Other effective modifications are an input-scaled noise schedule, x-prediction and using tanh as a final activation function.

While our approach does not yet surpass leading mask-based universal models in general foundation settings [2, 10, 11], it narrows the gap and offers capabilities those models lack: principled ambiguity modeling and sample-based exploration of plausible masks. Given the progress of diffusion segmenters like UniGS [8], we see a possible path forward: combine large-scale pretraining with analog bits and as many as the other proposed model improvements. Another path which might be more useful in practice is to integrate promptable conditioning combined with classifier-free guidance. If successful, generative universal segmenters could prove to be competitive models that remain both agnostic and holistic.

## 7    Acknowledgements

This work was supported by Danish Data Science Academy, which is funded by the Novo Nordisk Foundation (NNF21SA0069429) and VILLUM FONDEN (40516).

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
