# OpenReview forum: "Towards Agnostic and Holistic Universal Image Segmentation with Bit Diffusion"
_NLDL.org/2026/Conference — NLDL 2026 Poster_

### Official Review · Reviewer_NLoe · 2025-10-01
**Review of NLDL submission 46**

**Rating:** 4
**Confidence:** 1

**Summary:**

**Summary**

This paper introduces a diffusion-based approach for image segmentation. In contrast to traditional mask-based approaches, the proposed method can consider the entire image. The authors adapted three techniques from the literature to improve the performance of the framework, with experiments validating the performance of the approach.

**Strengths:**

**Strength**

I'm not familiar with the field of generative models and diffusion, but the paper seems well-written with a clear motivation.

**Weaknesses:**

**Weaknesses**

The major contribution of the paper appears to be the synthesis of several existing ideas from the literature, adapting them to the task of diffusion-based image segmentation. However, as mentioned in the conclusion, the performance of the proposed approach "does not yet surpass leading mask-based universal models in general settings". It would be better if the authors could add more comments on the types of tasks where the diffusion-based framework is most likely to be advantageous.

**Minor issues**

1. Line 28

   "you specialize" is not formal

2. Line 160

   $x_t$ => $\mathbf{x}_t$

3. Line 192

   enable => enables

4. Line 239

   What's the motivation for making the noise schedule harder?

**Justification:**

I'm not familiar with the field, but the paper appears well-written and supported by experiments.

---

> ### Author Rebuttal · Authors · 2025-10-22
>
> Thank you for you review. We try to address the weaknesses below:
>
>
> - It would be better if the authors could add more comments on the types of tasks where the diffusion-based framework is most likely to be advantageous.
>
> There are a few comments in the discussion paragraph, lines 443-455, but we will try to expand it.
>
>
> - Line 28. "you specialize" is not formal
>
> Thank you, we will rephrase:
>
> "Conversely, in narrow image segmentation, you specialize in a single dataset or task, such as brain tumor segmentation."
>
> $=>$
>
> "Conversely, narrow image segmentation is characterized by specializing on a single dataset or task, such as brain tumor segmentation."
>
> - Line 160. $x_t=>\textbf{x}_t$
>
> Thanks, will change it.
>
> - Line 192. enable $=>$ enables
>
> I think it should still be enable since it refers to "modification" and not "diffusion models".
>
> - Line 239. What's the motivation for making the noise schedule harder?
>
> With more noise (harder schedule), the model is forced to consider the image instead of predicting based solely on the latent sample, alleviating the problem displayed in Figure 2 (that you can guess the label without looking at the image). The connection between a harder noise schedule and this effect should be made clearer, so we will add a sentence explicitly stating this.

---

### Official Review · Reviewer_6oXo · 2025-10-02
**The review of "Towards Agnostic and Holistic Universal Image Segmentation with Bit Diffusion"**

**Rating:** 4
**Confidence:** 4

**Summary:**

This paper introduces a diffusion-based framework for universal image segmentation, tailored for discrete label domains. Key contributions include analog bit encoding, location-aware palette (LAP), input-scaled noise scheduling, tanh activation, and x-prediction. Experiments show that this method has certain effectiveness.

**Strengths:**

1. Provides a systematic adaptation of diffusion models for discrete segmentation using analog bit encoding with LAP.

2. Each modification (noise scheduling, tanh activation, x-prediction) is theoretically justified and empirically validated.

**Weaknesses:**

1. Evaluation is restricted to Entity-Seg only, lacking cross-domain validation (e.g., medical or remote sensing).

2. No direct quantitative comparison with recent strong baselines like MaskFormer[1]/AUCSeg[2].

3. Missing analysis of GPU time, memory footprint, or scalability for large-scale deployment.

4. While multiple improvements are proposed, the paper does not sufficiently analyze their interdependencies or independent contributions.

[1] Per-Pixel Classification is Not All You Need for Semantic Segmentation. NeurIPS 2021.

[2] Aucseg: Auc-oriented pixel-level long-tail semantic segmentation. NeurIPS 2024.

**Justification:**

My recommendation for acceptance is based on the fact that the proposed method and conclusions are both exciting and promising. That said, there are still areas that require improvement, such as providing more comprehensive baselines for comparison and conducting more detailed experimental analyses.

---

> ### Author Rebuttal · Authors · 2025-10-22
>
> Thank you very much for your review. We try to address the comments and weaknesses below:
>
> - Evaluation is restricted to Entity-Seg only, lacking cross-domain validation (e.g., medical or remote sensing).
>
> We considered many datasets and also trained earlier models on large cohorts of many datasets including medical/photos. In the end, we don't think it adds much to the article since the model isn't competitive with SOTA. Because of comparisons and due to the ease of explaining our data pipeline we decided to just use entityseg as it is varied and high quality, especially compared to ADE20K, COCO, etc.
>
>
> - No direct quantitative comparison with recent strong baselines like MaskFormer[1]/AUCSeg[2].
>
> Some other reviewers also found this lacking. We already made these tables, but left them out in the interest of space. We will add a table with comparisons to Mask2Former and Cropformer.
>
>
> - Missing analysis of GPU time, memory footprint, or scalability for large-scale deployment.
>
> We think it would definitely benefit the paper to have such an analysis, but in the interest of space and because the research is exploratory we decided to skip it.
>
>
> - While multiple improvements are proposed, the paper does not sufficiently analyze their interdependencies or independent contributions.
>
> This is a good point, we decided to basically go with a linear addition of method A, B, C and seeing the performance jump at each step. This order was the same as the one we discovered the improvements in. Not included in the paper are some experiments where we tried adding improvements in a different order, say C, A, B.
> Sometimes the improvements per model addition would change just a little bit and other times very significantly. As an example, training models without input scaling would yield almost degenerative models that would give nonsensical or just low-quality results. Asking whether bit diffusion+tanh or LAP is best to add when no input scaling was present was therefore not very insightful.
> In the end, regardless of order, adding all 3 improvements was always best. We decided trying to convey this would be a mess, but based on your comment, we can see it has value to the reader who might think "I don't need A, we can just go with B and C".
>
> We will add a discussion paragraph describing the interdependence.

---

### Official Review · Reviewer_xooQ · 2025-10-05
**Review Paper 46**

**Rating:** 2
**Confidence:** 3
**Final Rating:** 2
**Final Confidence:** 3

**Summary:**

This paper presents a diffusion-based framework for universal image segmentation, enabling holistic segmentation without relying on mask-based approaches. They introduce several key adaptations for applying diffusion models in this discrete setting: a location-aware palette with 2D gray code ordering, a final tanh activation for discrete outputs, and a sigmoid-weighted loss for parameter optimization, with x-prediction chosen as the preferred strategy. While the current model does not yet surpass state-of-the-art mask-based architectures, it narrows the performance gap and offers unique capabilities, such as principled modeling of segmentation ambiguity. All models were trained from scratch, and they anticipate that integrating the improvements with large-scale pretraining or promptable conditioning could yield highly competitive results.

**Strengths:**

- The topic addressed by the authors is important, and it seems reasonable to use diffusion models.
- The dataset is appropriate for the topic, as well as the evaluation metrics.
- The numerous ablation studies provide valuable insights and support the method. The individual steps of the method seem appropriate.
- The authors highlight many opportunities for further development of the method and next steps.

**Weaknesses:**

- Mask-RCNN is not a universal model, but rather an instance segmentation model, as it is trained on specific datasets and only performs instance segmentation. So Mask-RCNN is not a foundation model either, as stated in the introduction.
- I am missing  references in related work, especially for the statements in lines 103-114.
- pix2seq-D is cited as a recent diffusion-based method, but the work dates back to 2022. The related work section seems very incomplete to me, with only one paper cited that deals with diffusion-based panoptic segmentation and only one that deals with universal image segmentation. For example, these papers could also be cited: "DFormer: Diffusion-guided Transformer for Universal Image Segmentation" Wang et al, "Open-Vocabulary Panoptic Segmentation With Text-to-Image Diffusion Models" Xu et al

- It seems that all three contributions have already been applied to image segmentation, so what is new here, the plugging together? The novelty should be explained in more detail.
- Compared to pix2seq-D, the added value of the paper seems to be the use of the location-aware palette, and compared to UniGS, it is mainly the training from scratch. At this point, the authors' method should be compared in more detail.

- "We employ a convolutional neural network (CNN) with an attention mechanism" Why not use transformers directly? What are the advantages of this hybrid architecture?
- The comparisons section also seems more like an ablation. I miss the comparison with other methods, such as UniGS.
- The authors state that their method does not outperform mask-based methods such as MaskFormer, but no numbers are provided for comparison, not even with other diffusion-based approaches.

- I find the abstract very complicated to read, very overloaded with information and limitations.
- Alpha and sigma in equation 1 should be explained.
- In line 278 "(see supplementary material for details)" no supplement there
- There are many hyperparameters; a brief overview in section 4.1 could be helpful.
- The references are cited uncleanly. For some papers, only arxiv is cited instead of the conference version, for example "UniGS: Unified
Representation for Image Generation and Segmentation" CVPR 2024 or "A generalist framework for panoptic segmentation of images and videos" ICCV 2023

**Final Justification:**

After reading the author's rebuttal, I am somewhat more positive. I am maintaining my score of 2, as I cannot give a clear acceptance, but if the other reviewers see it more positively, I will not vote against it.

**Justification:**

I like the authors' idea and think it is an important area of research, but the related work section is outdated/incomplete and there is no comparison with SOTA methods. The paper seems like a small study (many ablations), but the authors' own novelties are unclear. In addition, there are errors/ambiguities in the paper that should be corrected.

---

> ### Author Rebuttal · Authors · 2025-10-22
>
> Thank you for your review and time. Here are a few comments on addressing the weaknesses from your review:
>
> - Mask-RCNN is not a universal model, but rather an instance segmentation model, as it is trained on specific datasets and only performs instance segmentation. So Mask-RCNN is not a foundation model either, as stated in the introduction.
>
> That is an error on our part, and it will be fixed. It was written since the use case of Mask-RCNN trained on e.g. COCO was quite similarly to an agnostic segmentation tool like SAM is today, causing our confusion.
>
>
> - I am missing references in related work, especially for the statements in lines 103-114.
>
> The start of this paragraph is essentially just our observations on the agnostic segmentation field. It was put in the related works section as it motivates the choices and smoothly transitions from one type of model to the next. We see that it is problematic that this is not addressed, especially in a related works section. We suggest a new start for the paragraph reading: "We observe that in order to train a deterministic universal model, a concession has to be made. Mask-based models are ..."
>
> - pix2seq-D is cited as a recent diffusion-based method, but the work dates back to 2022. The related work section seems very incomplete to me, with only one paper cited that deals with diffusion-based panoptic segmentation and only one that deals with universal image segmentation. For example, these papers could also be cited: "DFormer: Diffusion-guided Transformer for Universal Image Segmentation" Wang et al, "Open-Vocabulary Panoptic Segmentation With Text-to-Image Diffusion Models" Xu et al
>
> We will remove the word "recent". The DFormer paper is not as comparable as the Pix2seq-D paper, since it is not holistic. They followed a mask-based approach similar to mask2former, DETR.
> For Xu et al., the model is not generative and essentially just uses the features that the diffusion model extracts as an image encoder.
>
> Our related works section could definently be improved, and we will add a some more sentences to describe the landscape of research more accurately - including your mentioned papers.
>
>
> - It seems that all three contributions have already been applied to image segmentation, so what is new here, the plugging together? The novelty should be explained in more detail.
>
> Combining them, yes, but concretely adding a Tanh activation and optimizing the LAP by adding 2D gray codes. We will make the novelty more explicit in the "our contributions" list.
>
>
> - "We employ a convolutional neural network (CNN) with an attention mechanism" Why not use transformers directly? What are the advantages of this hybrid architecture?
>
> Because it is easier to train and work with and if the performance gap was smaller, and there was a big benefit in trying to gain a few more percentages, we would've used transformers.
>
>
> - Compared to pix2seq-D, the added value of the paper seems to be the use of the location-aware palette, and compared to UniGS, it is mainly the training from scratch. At this point, the authors' method should be compared in more detail.
> - The comparisons section also seems more like an ablation. I miss the comparison with other methods, such as UniGS.
> - The authors state that their method does not outperform mask-based methods such as MaskFormer, but no numbers are provided for comparison, not even with other diffusion-based approaches.
>
> We already made comparisons to Mask2Former and Cropformer but left them out in the interest of space, they will be added back in.
>
>
> - I find the abstract very complicated to read, very overloaded with information and limitations.
>
> Thank you for your feedback. We'll revise the abstract to try and make it clearer.
>
> - Alpha and sigma in equation 1 should be explained.
>
> We will add the following explanation sentence: "The functions $\alpha(t)$ and $\sigma(t)$ are the data and noise coefficients, respectively."
>
>
> - In line 278 "(see supplementary material for details)" no supplement there
>
> The supplementary materials were not supposed to be included in the submission on the submission platform. (I think it's also not supposed to be considered for the review either).
>
>
> - There are many hyperparameters; a brief overview in section 4.1 could be helpful.
>
> We will a parameter overview table in the supplementary material.
>
>
> - The references are cited uncleanly. For some papers, only arxiv is cited instead of the conference version, for example, "UniGS: Unified Representation for Image Generation and Segmentation" CVPR 2024 or "A generalist framework for panoptic segmentation of images and videos" ICCV 2023
>
> This will be fixed.
>
> Thank you for your careful review and constructive feedback, and we hope we were able to convince you to change your decision.

---

### Official Review · Reviewer_b332 · 2025-10-06
**Review of "Towards Agnostic and Holistic Universal Image Segmentation with Bit Diffusion"**

**Rating:** 2
**Confidence:** 2

**Summary:**

In this article the author presents an approach to perform universal image segmentation which deals with the two competing problems:
Agnostic behavior, which is the ability for a model to segment an object without relying on fixed labels, and holistic image views, which in this setting means the model consider the whole image when producing segmentation.

The methods section goes into the basics of diffusion models which works well going further into Bit Diffusion which is a core part of the article. Bit diffusion consists of switching out one-hot encoded labels with a bit system, so instead of having a 64 length one-hot encoded vector, you have a bit-representation of the class instead. This then comes out as a length 6 vector instead. The author also add that instead of using 0 as the negative bit, they use -1 instead. This works well with the tanh activation function which is used as the final activation function in the network.
The final part of the Bit Diffusion section is a bit confusing, as it's not made clear why considering each bit separately would be problematic, yet the tone of the section makes it seem like it is. Maybe this is obvious to those who are more familiar with the subject than me, but to me this part reads a bit weird and is confusing.
In addition to Bit Diffusion the authors include noise scheduling and input scaling to better adopt the noises strength and occurence based on their images.
Lastly for the methods, the authors include a Location-aware Palette (LAP) in order do deal with cases where the agnostic model segments every little object separately. This method divides the image into a grid and assignes rules for what happens to segment centroids which share grids among other benefits.


The authors do several experiments.
First they compare their proposed model with with models trained using RGB and onehot encodings. The authors show that in both cases where they show that in the case of using LAP and not using LAP their proposed use of bit diffusion outperformed the traditional oneshot and the RGB embedding scheme. The improvements are measured in adjusted rand index (ARI) and IoU, and show the mean performance of the different models. Given that these results are the mean I'd expect to see standard deviation / variance results too. These are not present. The result presented does paint a very good picture of the authors proposed method in terms of performance.
The authors do show some extra plots, especially Figure 3, which shows how the different metrics change by the number of classes. We see that the proposed method can go up to around 256 classes before a performance drop versus the onehot encoding which drops off around 16. I do note that the resolution for these performance plots aren't great, but they show the trend they set out to show.

The next experiments presented some results where the authors are doing some sampling, but it not completely clear to me exactly what is being sampled. Is it noise? Or is it traningdata in some way? I'l' not make more comments on the second half of section 4.2 as I'm not completely sure what is being referred to.



Section 4.3 explores the effect of different loss weights and prediction types on model performance. The authors show that in all chases the sigmoid loss weights come out on top. They also find that predicting the noise (epsilon prediction) is slighty better than image prediction (x prediction) on average. They do however find that models trained om predicting the noise it often fails to predict all the noise properly.

In addition the authors show their model performance on different values of b for their input scaling. They conclude that a value of b=0.1 is clode to optimal for their application, yet figure 9 makes it seem like a value of b=0.03 might be better.

For section 5, the discussion section, the authors bring up a few interesting points. From what I've seen in the article I agree on their LAP observations and how it affects the network.

**Strengths:**

This section presents the problem and to-be solution in a clear and concise manner. Key terms such as agnostic behavior and and holistic view is clearly defined. The contributions of the author in also clearly defined.
The motivation to as why I this article is important comes out naturally from the proposed problem and solution.

The methods section clearly defines the fundamentals this article builds upon, and is generally easy to follow. The bit diffusion is explained easily, and LAP is explained well.

The experimental setup is explained well, and all the experiments serves a purpose. They showcase that the three main contributions listed in the introduction have been achieved.

**Weaknesses:**

Not sure why the results in Table 1, Table 2, and Figure 7 shows mean performance metrics but not standard deviation. This should have standard deviations added to the results.

This article lacks experiments comparing itself to other models / methods to showcase its performance relative to those methods and showcasing it does truly move towards fulfilling the goal of moving towards agnostic behavior and holistic view of images.

Some terms, such as sampling in section 4.1, is not clear as to what they are referring to.

Some smaller stuff:
In section 4.1 , the "attn-UNet" is missing a source. I looked up potential candidates and it's probably the model proposed in Oktay et.al 2018, but this isn't clear.

**Justification:**

I am very on the fence about this article.
The presented problem is clear and seems like one which is important to solve. The article is generally clear on its contents. The introduction and methods section are well written, and easy to read.

I am however not completely sold on the experiment section. What is there is good, but it needs more and half of section 4.1 needs to be made clearer. I would also like to see a comparison to other models which makes the argument that adapting input scaled noise scheduling, bit diffusion, and LAP actually leads to a more general and universal model is true. The results doesn't need to show it's the best, only that these additions somewhat achieve what you say they do.

If the other reviewers say this article is an accept however, I will not argue against that.
I believe you have something very interesting in this article, but it just needs a bit more in the experiment section to put this method into a larger context.
As it is now however, I'm unfortunately leaning towards reject.

---

> ### Author Rebuttal · Authors · 2025-10-22
>
> Thank you for your review and time. Here are a few comments on addressing the weaknesses from your review, as well as a some of your other comments:
>
> - The final part of the Bit Diffusion section is a bit confusing, as it's not made clear why considering each bit separately would be problematic, yet the tone of the section makes it seem like it is. Maybe this is obvious to those who are more familiar with the subject than me, but to me this part reads a bit weird and is confusing.
>
> In the submitted version the downside is touched upon in the Location-Aware-Palette section, but we now realize that the point is unclear. In the revised version we will try to make this connection clearer so the point doesn't stand so loose.
>
> The point is that compressing e.g. a 64-dimensional probability space into a 6-dimensional one reduces representational power because correlations between bits are ignored. In reality, the bits are correlated and LAP reduces the the loss in representation.
>
>
> - I do note that the resolution for these performance plots aren't great
>
> There was a restriction, since the number of bits dictates total number of classes to be a power of 2 and the LAP restricts the number of classes to be a square number. Therefore the numbers 4,16,64,256,1024 were chosen for the plot. Extending the methods to make other class numbers would be possible but relatively cumbersome, so we stuck with these values.
>
>
> - The next experiments presented some results where the authors are doing some sampling, but it not completely clear to me exactly what is being sampled. Is it noise? Or is it traningdata in some way?
> - Some terms, such as sampling in section 4.1, is not clear as to what they are referring to.
>
>
> It is the noise, initializing the diffusion model with randomly sampled noise maps to obtain probabilistic results (multiple predictions per image). We will make this clearer in the start of the section, as just using the word sampling is too vague.
>
>
> - Not sure why the results in Table 1, Table 2, and Figure 7 shows mean performance metrics but not standard deviation. This should have standard deviations added to the results.
>
> We will add these in the revised version.
>
>
> - This article lacks experiments comparing itself to other models / methods to showcase its performance relative to those methods and showcasing it does truly move towards fulfilling the goal of moving towards agnostic behavior and holistic view of images.
>
> Some other reviewers also found this lacking. We already made these tables but left them out in the interest of space. We will add a table with comparisons to Mask2Former and Cropformer.
>
>
> - Some smaller stuff: In section 4.1 , the "attn-UNet" is missing a source. I looked up potential candidates and it's probably the model proposed in Oktay et.al 2018, but this isn't clear.
>
> You're right we're missing a reference for the architecture. It should be [25] Hoogemoom et al. Simpler Diffusion (SiD2). We will add this.
>
> Thank you for your careful review and constructive feedback, and we hope we were able to convince you to change your decision.

---

### Official Review · Reviewer_CnWb · 2025-10-07
**Review for Paper 46**

**Rating:** 5
**Confidence:** 4

**Summary:**

The authors propose a diffusion-based approach for universal segmentation utilizing analog bit diffusion, location-aware palettes (LAP), and input scaling.  Each of these concepts is thoroughly explained in the methods.  The segmentation models are tested and evaluated with ARI and IoU comparing analog bit diffusion to RGB and onehot encoding, while also comparing models with LAP to models without LAP; these tests show that analog bits with LAP perform the best.  Ablation studies are then described which isolate the best setup for LAP, data prediction type, and input scaling for the given task with Entity-Seg dataset.  The authors show their proposed setup has promising results but does not currently rival other state-of-the-art mask-based segmentation models.  The authors postulate this gap likely corresponds to differences in scale and the proposed methods may be beneficial to larger scale models.

**Strengths:**

- Introduction and related works sections are concise yet comprehensive.
- Very clear and well-written description of the proposed methodology.
- Clearly addresses both strengths and weaknesses of the proposed methods.
- Experiments and ablation studies are well designed to evaluate the proposed model.

**Weaknesses:**

- Performance evaluation values such as ARI and IoU are stated to be mean values, but variance of the estimates is not listed, nor are the number of replicates performed for each test.
- It is stated that the model is not competitive with larger-scale mask-based segmentation models, but no explicit and numerical results are shown before discussing this gap in performance.

**Justification:**

The authors clearly describe and test an architecture that shows promise for improving universal segmentation methods.  They clearly describe benefits of their proposed method while addressing shortcomings and propose several avenues for future research could further improve on their own work as well as related works.

---

> ### Author Rebuttal · Authors · 2025-10-22
>
> Thank you very much for your review. Here are our comments on addressing the weaknesses:
> - Performance evaluation values such as ARI and IoU are stated to be mean values, but variance of the estimates is not listed, nor are the number of replicates performed for each test.
>
> We evaluated the method on 10\% out of 33k samples, so just around 3300 samples. We used one evaluation per sample. We will add this explanation, as well as $\pm std$ to the revised version.
> - It is stated that the model is not competitive with larger-scale mask-based segmentation models, but no explicit and numerical results are shown before discussing this gap in performance.
>
> Some other reviewers also found this lacking. We already made these tables but left them out in the interest of space. We will add a table with comparisons to Mask2Former and Cropformer.

---

### Meta-Review · Area_Chair_idiY · 2025-10-28

**Recommendation:** Accept (Poster)
**Confidence:** 3

**Metareview:**

This paper studies image segmentation through generative learning. Specifically, a diffusion framework is proposed for universal image segmentation, bypassing default mask-based frameworks. This paper received 5 reviews with mixed opinions. All reviewers agree that the approach is interesting and the introduction and methodology are clearly presented. The reviewers also highlight a few concerns, such as a lack of comparisons, explanations of experiments, and missing related work. The rebuttal partly addresses these points. Overall, the AC believes that the suggestions made by the reviewers improve the paper and the authors are open about the overall performance of the model and the gap to the SOTA. The AC recommends acceptance and urges the authors to update the paper with the suggestions by the reviewers, as well as an explanation about the overall performance and the gap to the highest performing models in the field.

---

### Decision · Program_Chairs · 2025-11-05

**Decision:**

Accept (Poster)

**Comment:**

We recommend a poster presentation given the AC and reviewers recommendations.